

# Potential Impact of Carbonate Chemistry Change (pCO2) on Krill and Krill-based Food chain in the Southern Ocean with emphasis on Embryogenesis of Antarctic krill.

**Dr. Robert Y. George**
President, George Institute for Biodiversity and Sustainability (GIBS)
Wake Forest, North Carolina, USA
www.GIBSconservation.org
Correspondence to: Dr. Robert Y. George (georgeryt@cs.com)

**ABSTRACT**

In the Southern Ocean, it is still not certain that overall krill biomass may decline because of drastic increase in pCO2, and consequent decline in pH. However, there is evidence that ecological vacuums created by krill population collapses caused by ecosystem shifts in Western Antarctic Peninsula (WAP) region led to replacement of Antarctic krill *Euphausia superba* by soft-bodied salps *Salpa thomsoni*. There is yet another questionable hypothesis that by the end of 21st century, ocean acidification stress, coupled with thermal increase, may synergistically induce physiologically critical stress to Antarctic krill in some areas of the Southern Ocean, egg development of krill may drastically decrease and in the 23rd century krill may even become extinct. I have earlier reported on normal krill egg development in relation to thermal change and high pressure (George and Stromberg, 1985). Recent experiments on krill development under different pCO2 conditions by Kawaguchi et al, (2011, 2013) suggest that we may witness 20 to 70 % reduction in Antarctic Krill by 2100 as direct consequence of pH decline. Such a scenario may lead to demise of krill-eating top-predators like baleen whales, seals and different species of Antarctic penguin populations. We now know that Adelaide penguins are decreasing in Bransfield Strait region off of the Western Antarctic Peninsula but increasing in Ross Sea region. Such a shift in breeding colonies moving from northern to southern WAP region and Ross Sea areas is not attributed to any decline in krill biomass but recent decadal melting of sea-ice as documented by remote sensing (George and Hayden, 2017). In this paper the main focus revolves around implications of changing chemistry of the Southern Ocean caused by absorption of anthropogenic carbon dioxide.

## 1 INTRODUCTION

Our knowledge on the physiological responses of Antarctic krill *Euphausia superba* to increased pCO2 is meager. However, Saba et al. (2012) reported from experimental studies on *E. superba* that sufficiently elevated CO2 concentrations could alter internal acid base balance, compromising homeostatic regulation and disrupting internal systems ranging from oxygen transport to ion balance. Perturbation experiments with *E. superba* under elevated CO2 (672 ppm) ingestion rates of krill averaged 78 microgram C per individual, per day, this is 3.5 times higher than krill ingestion rates at present day CO2 concentrations. Rates of ammonia excretion by krill were 1.5 times higher. George and Fields (1984) reported that ammonia excretion rate in freshly captured krill is exceedingly high with peak levels reaching as much as 260 micrograms NH3/g.dry weight/hr. George (1985) discussed the implications of ammonia input to the ambient environment and its association with areas of large krill swarms, with over 1000 krill per cubic meter, both in the areas northwest of Elephant Island and in areas in Bransfield Strait in the West Antarctic Peninsular (WAP) region. Input of ammonia from krill is a significant mechanism to induce diatom bloom and therefore high primary production. In the summer months around South Georgia (Clark and Moriss, 1983), an adult krill grows as much as 2 mm per week, almost about 1 cm per month. Male adult krill has a daily energy intake of 5% of body weight per day. For a female krill the daily energy uptake can be as much a 6% of body weight.



Climate warming and sea ice disintegration are well known in the Western Antarctic Peninsular (WAP) region. The most interesting evidence is the shift in ecosystem with greater abundance of soft-bodied Salps, *Salpa thomsoni* (plankonic Tunicates), replacing the vacuum
left by decline of krill (Loeb and Santora, 2012). In the WAP region, net primary productivity also declined during the past three decades, associated with change in phytoplankton community composition, potentially impacting negatively on krill and positively on salp grazing efficiency, Schoeffield et al., 2010). Decrease in availability of krill as food around South Georgia also influenced predator abundance, with reference to penguins, seals and whales (Hinke et el.,
2017). However, the response was species-specific and certain populations of the Antarctic fur seal (Christensen, 2006) and humpback whales *Megaptera novaezeandiae* (Nicol et al., 2008) have increased considerably over the past decades. Flores et.al. (2012) have warned that by "the end of 21st century, levels of warming and ocean acidification may reach physiologically critical levels in some areas in the Southern Ocean particularly for early developmental stages
of krill *Euphausia superba."*

Trivelpiece et al. (2011) clarified the misconception that ice-loving Adelie penguins declined in WAP region and Scotia Sea and Ice-avoiding chinstrap penguins increased with decrease of krill biomass as a consequence of declining winter ice habitats due to warming
since early 1980s. Emslie et al. (2007) also reported on a 45,000 yr record of Adelie penguins and climate change in the Ross Sea, Antarctica. A huge threat to krill in the Southern Ocean may not come from the rapid warming off of the southern WAP region, but it is likely to come from carbonate chemistry changes in the water column all around the Antarctic continent in the Southern Ocean. In fact, we have still not perfected the methodology to measure directly partial
pressure of pCO2. We know that high latitude ecosystems absorb more anthropogenic carbon dioxide because of the low temperature.

Recent advancement in grasping pCO2 threats in Southern Ocean come from use of data derived from Argo floats. Williams et al. (2016) discussed the importance of using empirical
algorithms to estimate water column pH in the Southern Ocean. Sensors on Argo floats include measurement of salinity, pressure, nitrate, and dissolved oxygen in the Pacific Sector of the Southern Ocean. Algorithms were applied to Southern Ocean Carbon Climate Observations and Modeling (SOCCOM) to discern biogeochemical seasonal cycles.

We have often singled out one parameter and asked the question how temperature increase in a low temperature stenothermal ecosystem will influence at the cellular, organism, species, community or ecosystem levels. Realistically we need to look at climate change impact at the multiple influence of pH decrease, temperature increase and drop in dissolved oxygen causing hypoxia. Reum et al. (2016) synthesized a large carbonate chemistry data-set that consists of
carbonate chemistry, temperature, and oxygen measurements from multiple moorings and ship-based sampling campaigns from the California Current Ecosystem (CCE), and includes fjord and tidal estuaries and open coastal waters. They evaluated patterns of pCO2 variability and highlighted important co-variations between pCO2, temperature, and oxygen. They compared environmental pCO2–temperature measurements with conditions maintained in OA experiments
that used organisms from the CCE. By drawing such comparisons, researchers can gain insight into the ecological relevance of previously published OA experiments, but also identify species or life history stages that may already be influenced by contemporary carbonate chemistry conditions.

Undoubtedly, atmospheric carbon dioxide keeps on increasing because of fossil fuel burning (Rhein et al, 2013), with 42 % of the excess carbon staying in the atmosphere and about 58 %



absorbed by the ocean and terrestrial biosphere. Models thus far revealed that the ocean has sequestered approximately 28 % of all anthropogenic CO2 and this has caused decrease in surface ocean pH of approximately 0.1 (Orr et al., 2005). The Southern Ocean plays a major role in the uptake and long-term storage of CO2 accounting for 40 % of the total CO2 sink
(Frolicher at al., 2015). The focus of this paper deals with how Antarctic krill will cope with the potential threat of pH decline in Southern Ocean in the coming decades, unto 2100.

## 1.1 KAWAGUCHI ET AL. 2011 HYPOTHESIS

Kawaguchi et al. (2011) exposed krill (*Euphausia superba*) embryos to three different experimental  pCO2 conditions, 380, 1000 and 2000 microatoms with the goal to determine the potential impact of increased ocean acidification in the Southern Ocean. They discovered that normal embryogenesis occurred at 280 and 1000 microatoms of pCO2 but at 2000 microatoms, 90 % of egg development was disrupted before gastrulation. In the opinion of this author,
Southern Ocean will never reach such high carbonate chemistry by 2100 even with atmospheric CO2 at 780 ppm or seawater at 1400 microatoms pCo2. It is important that we establish the influence of the synergy between increasing temperature and decreasing pH (increasing pCO2) in the Southern Ocean. There is also evidence that in the Western Antarctic Peninsular (WAP) surface water temperature increased as much as 2 C and sea-ice melted over the recent
decades (George and Hayden (2017).

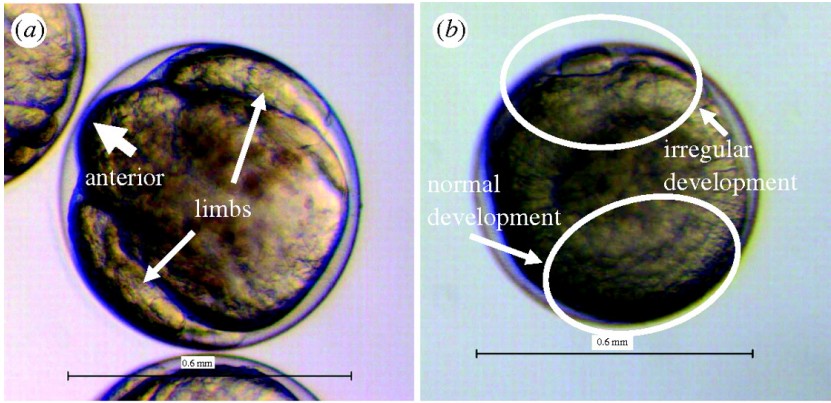

Fig. 1. Left: Normal embryogenesis leading to twitching limb-bud embryos unto 1000 microatoms. Right. Abnormal embryos at 2000 microatoms pCO2.


Kawaguchi et al. (2011) discovered that embryos reared under 380 microatom (current pCO2 conditions) to 1000 microatoms developed into limb bud stage (see figure below) but at 2000 microatoms, embryonic development was disrupted during gastrulation with a large portion of ectoderm appearing irregular.





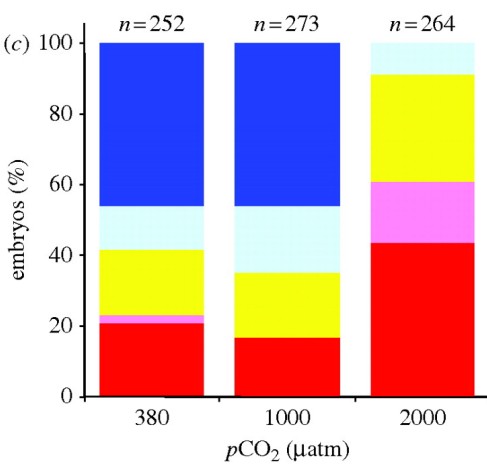

Fig. 2. Krill embryogenesis under
different CO2 conditions

## 1.2 KAWAGUCHI ET AL., 2013 FINDINGS

 As pointed out by Kawaguchi el al, (2013) it is not the surface but the deep depths of the water column that the low pH acidic stress will be experienced by the descending developing eggs of krill, as illustrated in the figure below. Depths between 100 and 1000 meters are the vulnerable zone with high pCO2.

25

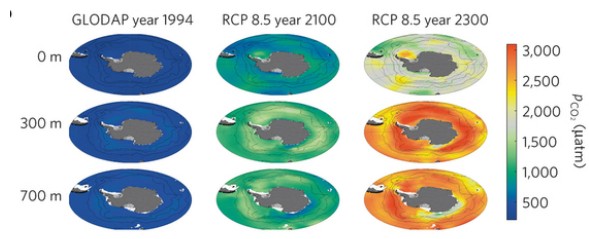

Fig. 3.   Vulnerable Zone (red) Southern Ocean
in 3 Different Scenario

These authors expressed concern about ocean acidification and expanding krill fisheries and their dependent top predators – baleen whales, seals and penguins (Flores et al., 2012) with Southern Ocean reaching very high (lethal) pCO2 conditions in 2300. Krill embryos tend to sink down below 100 meters within 12 hrs after spawning and can reach down to 1000 meters before hatching. Risk maps for hatching success were generated (see below) for Weddell Sea, and by 2300 the entire Southern Ocean will become unsuitable for hatching. Obviously, we need predicted models on hatching success of krill embryos for the coming decades rather than risk maps projected for a distant future.

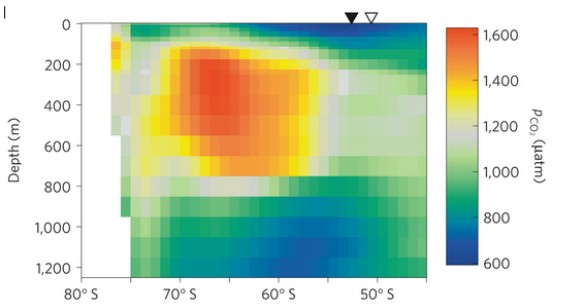



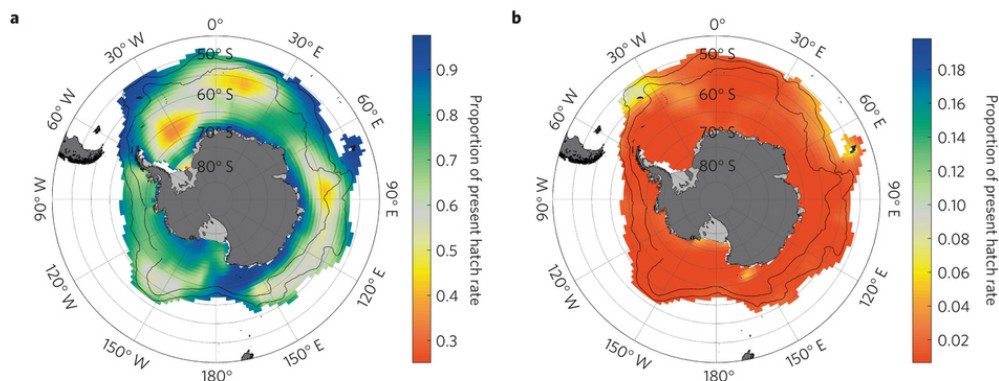

Fig. 4. Circumpolar risk-maps of krill hatching under projected future pCO2 levels,
figure on left 2100 and figure one right 2300.

## 2 GEORGE'S ORIGINAL DEFINITION OF KRILL EMBRYOGENESIS

     Fertilized eggs of *Euphausia superba* from gravid females, were subjected to electron microscopic examination (Dillaman, George and Stromberg, 1985). The center of the centrolecithal krill egg was filled with yolk droplets and vast numbers of mitochondria. This finding lends support to our hypothesis that krill eggs, while descending from surface depths to
deeper depths (600 to 1000 m or more), has enormous sources of metabolic energy to propel blastulation, gastrualtion and organogenesis. Amsler and George (1985) discovered that krill eggs during development exhibit an atypical model of metabolic pattern. Each egg is about 30 micograms and is composed of 37.5 % lipid, and during developmental descent, until nauplius stage, twice the protein resources are used than the lipid that is saved for buoyancy during the
ascending phase until the first feeding Calytopis I larval stage is reached.

### 2.1 MATERIAL AND METHODS

    The gravid female krill in healthy conditions were caught off Anvers Island in Palmer Peninsula
with a plankton net that was equipped with a 30 gallon acrylic chamber zipped to the cod end. Over a hundred healthy gravid female krill were transferred to cold aquaria kept at 1 C at Palmer station. Ten krill were selected and each kept in a half-gallon glass jar with a plastic perforated platform at the middle and eggs were collected below in a crucible that was then transferred to the pressure chamber described below.


### 2.3 EXPERIMENTAL SET-UP

### PRESSURE CHAMBER FOR MONITORING KRILL EMBRYOGENESIS FROM FERTILIZED EGG TO HATCHIG OF NAUPLIUS LARVA AT 100 ATM


The krill embryos, while sinking encounters increasing hydrostatic pressure, and 10 to 100 atm ambient pressure must be considered as an important factor in krill development. George and Stromberg (1985) used an experimental set-up (see figure below) to study krill embryology from cleavage, through blastula, gastrula, lib-bud and to hatching nauplius, establishing definition of
these embryological stages, as depicted in the figure below.




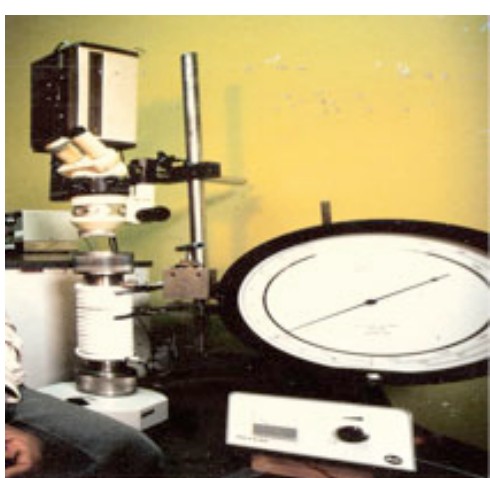
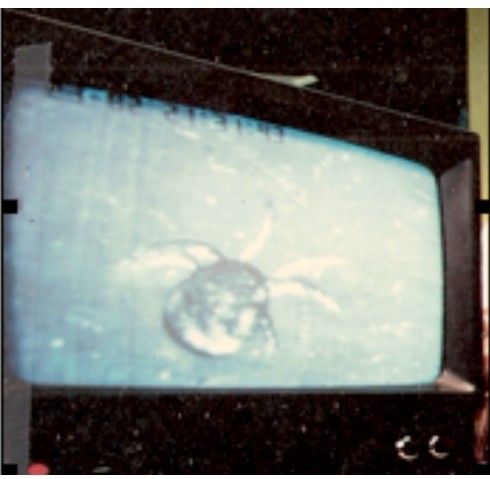

Fig. 5. left – Pressure chamber outfitted with Right - Newly hatched nauplius larva
Microscope to observe krill egg development at 100 atm in a simulated sinking
with a TV camera to videotape different rate in the laboratory in Palmer,
stages such as blastulation, gastrulation and limb-bud formation.

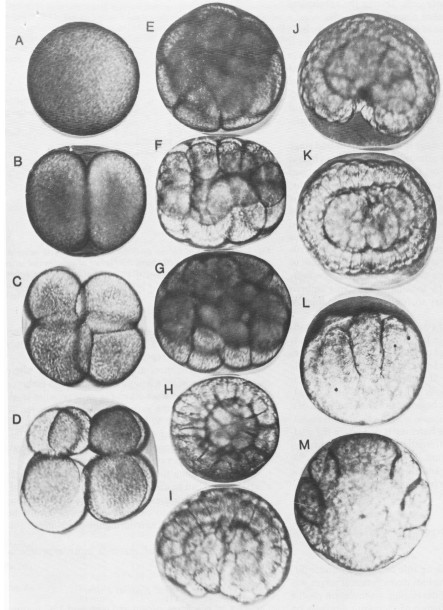

Fig. 6. Photos taken in the pressure chamber while krill eggs develop at 2 C.





### 3 IMPACT OF RECENT WARMING ON KRILL EGG DEVELOPMENT

George and Stromberg (1985) discovered that krill eggs failed to cleave at 4 C and therefore, the upper lethal level for krill is 4 C. However, we found that krill eggs kept at minus
0.5 C and plus 2 C, showed significant differences in the precise duration of embryogenesis from one stage to the other. For example, we found that it took 19 hrs at lower temperature for eggs to advance to a 4-celled stage but by increasing temperature by 2.5 C expedited cleavage a 4-celled stage was reached in 17 hrs. Likewise, the gastrula formation was attained in 58 hrs in higher experimental temperature but at minus 0.5 C it took 74 hrs. Gastrulation was advanced
by 20 hrs with an increase of 2.5 C. We also found that the duration for reaching limb-bud stage was reduced to 74 hr, as opposed to 97 hrs at  5 C. The development time from egg to hatching nauplius I larva was 150 hrs at 0.5 C but increasing temperature by 2.5 C reduced development time to 110 hrs.

The results presented above clearly indicate that warming in the Western Antarctic Peninsula is due to climate change definitely has brought about shrinkage in development time of krill embryos. As a consequence, normal hatching of krill nauplius larvae no longer happens at 1000 meter as in the years 1980 to 2010 but it possibly occurs at relatively shallower depths at 500 meters where we now see an ambient environment that has a reduced pH or pCO2
condition. This hypothesis calls for experiments in the future that takes into account temperature, pressure and pH or p CO2 simulated conditions to gain a good grasp of krill embryology in the light of a warming WAP region that is also undergoing concurrently carbonate chemistry change.


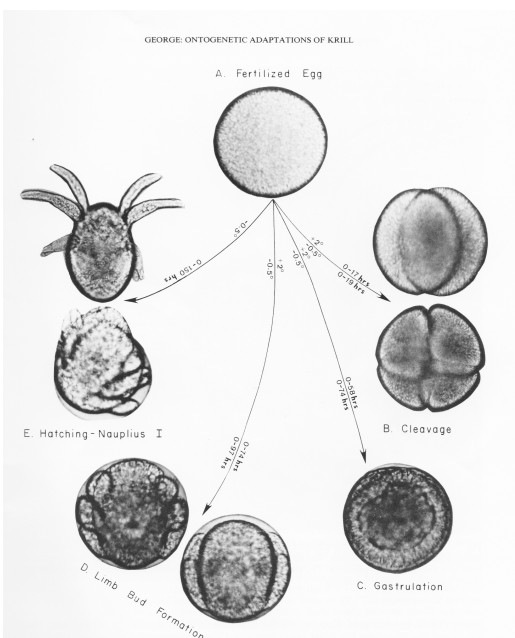

Fig. 7. Duration taken at minus 0.5 C vs. plus 2 C in krill egg development
of *Euphausia superba* at ambient pressure

### 4 KRILL IN WESTERN ANTARCTIC PENINSULA THEN (1983) and NOW (2017)

In the austral summer of 1983, three cruises aboard *R/V Hero* were conducted with the purpose of (A) collect gravid krill to transfer to cold aquaria in Palmer Research Station to obtain krill eggs to study their development in relation to subtle temperature increase and (B) determining the size frequency in schools of krill in the Bransfield Strait and several other locations in Palmer Archipelago (Fevolden and George, 1984).

Steinbeg et al (2014) reported on their long-term studies (1993 to 2013) For over 2 decades they observed an increase in
50  *Thyasonoessa macrura* over *E. superba* in 5 years abundance peaks, coinciding with



multivariate El Nino Southern Ocean Index and an association strongly influenced by primary production two years prior. Their study confirmed higher abundance of krill in higher ice conditions and on the higher abundance of thecosome pteropod abundance in lower ice conditions.

Bernard et al (2012) studied macrozooplankton abundance off of Western Antarctic Peninsula (WAP) in relation to the rapid climate change in this region of the Southern Ocean and they discovered that in the austral summers of 2009
and 2010 large blooms of *Salpa thompsoni* offshore while *E. superba* was the major grazer near
shore. In the Palmer Antarctic Long-term Experimental Research (LTER) they also found that *Salpa thompsoni* and pteropod *Limacina helicina* dominated and thereby proving clearly defined shifts in food chain dynamics in the WAP. Bernard and Steinberg (2013) explored the krill biomass and aggregations in WAP in relation to tidal cycles in a penguin foraging region. Tidal phase played a significant role in penguin foraging distance and krill aggregation was higher in
diurnal cycles. Saba et al (2014) discovered that krill abundance and biomass in the Western Antarctic Peninsula (WAP) region is simply a reflection of winter and spring controls.

Historically, the importance of Antarctic krill in the Southern Ocean marine ecosystem received new recognition at the First International Antarctic krill symposium in 1982 (George,
1984, ed.). The acoustic records of krill swarms off of Elephant Island and krill distribution in relation to biological and physical parameters, was the topic of a paper presented in the "Krill Workshop", organized by Prof. Gothilf Hempel in Bremerhaven in 1983 (George, 1983). Our knowledge on Antarctic krill in the Southern Ocean was further promoted by the 'Biomass Program' (El_Sayed, 1994). Subsequently, the second krill symposium consolidated our
knowledge on "Krill And The Unity of Biology" (Mengel and Nicol, 2000) at the 2nd KRILL symposium in Santa Cruz, California. Nevertheless, the significance of West Antarctic Peninsula region became a pivotal research focus area with the initiation of the Palmer Long-Term Ecological Research (LTER) by the Polar Program of the US National Science Foundation, with the goal of "investigating the oceanic, atmospheric and biogeochemical processes that results
from natural disturbances, environmental change and human impacts." The sampling grid (see figure below) consists of 180,000 sq. km surrounding Palmer Station.







Fig. 8. Depth integrated RSSI for 2015 Cruise Projection: GCS Deception
Island. Map by Nicholas Alcaraz.

The pH vertical profile in the WAP region exhibits definite change between austral summer
5    vs. austral winter. There is also a year-to-year change in pCO2 or pH condition in relation to
depth as shown in the figure below from data derived from LTER region. Surface pH is low at
8.07 in the months from April 2014 to April 2015 but pH level became low at surface water from
July 2015 onward. Depths greater than 400 m exhibit a low pH scenario as low as 7.98 pH.
Meaning this is where hatching of krill nauplius larvae occur, pCo2 conditions are high with
10   lower pH than at upper depths in the water-column between surface and 400 m.

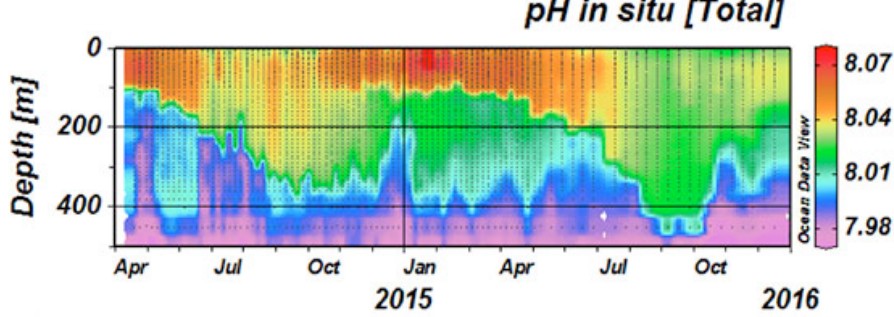

Fig. 9. pH profile in LTER study region in Western Antarctic Peninsula



There is also clear evidence from Kawaguchi et al.(2011) to conclude that the water column in the Scotia Sea where krill eggs hatch (400 to 1000 m) has 400 to 450 micoatoms pCO2 but in the Weddell Sea the pH is lower with water column pCOs ranging 500 to 580 micoataoms.

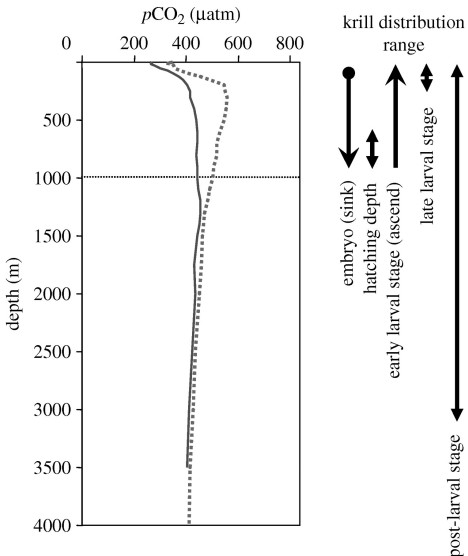

Fig. 10. Vertical Profiles of pCO2 in Scotia Sea Vs Weddell Sea in the Southern Ocean

George (1984) illustrated krill life cycle and in this diagram, given below, gives the impression that adult krill is confined to surface depths. However, recent reports from ROV studies suggest that Krill occurs at much deeper depths to feed during the winter months. Nicol (2006) also reported that there is difference in krill distribution between the summer and winter months. Further more, recent video observations confirm that *E. superba* leads a benthic life style at the sea-floor for mating at depths between 400 and 700 meters (Kawaguchi et al. 2011).



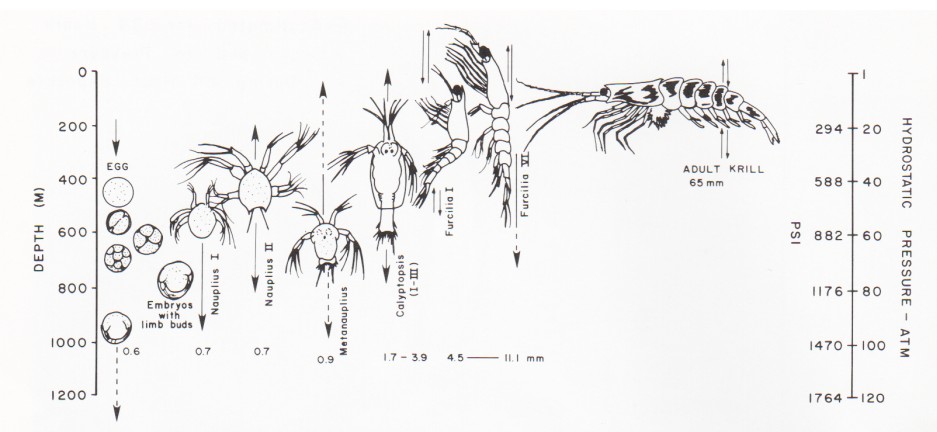

Fig. 11 Krill life cycle as illustrated by George (1984)

## 5 IS THERE A THREAT TO KRILL WITH 'ASH' SHOALING IN SOUTHERN OCEAN?

5      The answer to the question how krill life cycle will be impacted by the shoaling of the Aragonite Saturation Horizons (ASH) in the Southern Ocean. In a paper (Guinotte et al., 2006), that I coauthored with several prominent ocean acidification researchers (such as John Guinotte of USA, James Orr of France and Andrei Freiwald of Germany), we established that some taxa such as scleractinian corals react to changes in seawater chemistry, Ocean pH and calcium carbonate saturation as a consequence of influx of anthropogenic $CO_2$ to the atmosphere. There is evidence to prove that declining carbonate saturation inhibits the ability of marine organisms to build calcium carbonate skeletons that include exoskeleton of adults and larvae of crustaceans like krill. Here I have put forth a hypothesis that in the Southern Ocean, with atmospheric $CO_2$ at 440 ppm in 2020 and the near future, ASH depth in the West Antarctic Peninsular (WAP) region will be closer to 1000 m and therefore the life cycle of krill will not be influenced adversely (Fig. 12 A). Nevertheless, in 2100 ASH (Fig 13 B) ASH depth around WAP region is likely to reach 100 to 200 m with atmospheric $CO_2$ at 788 ppm. In such circumstances, krill embryogenesis can be in peril unless we take immediate mitigation actions to curtail current rate of $CO_2$ emissions to the atmosphere. In essence, this scenario compels actions to make paradigm shift from fossil fuel to renewable energy within the next two decades.

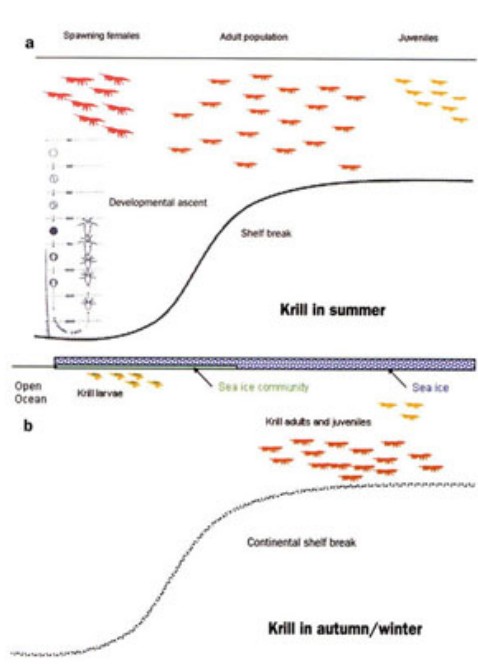

Fig. 12. Krill Distribution in the austral summer vs. winter (after Nicol, 2006)

35





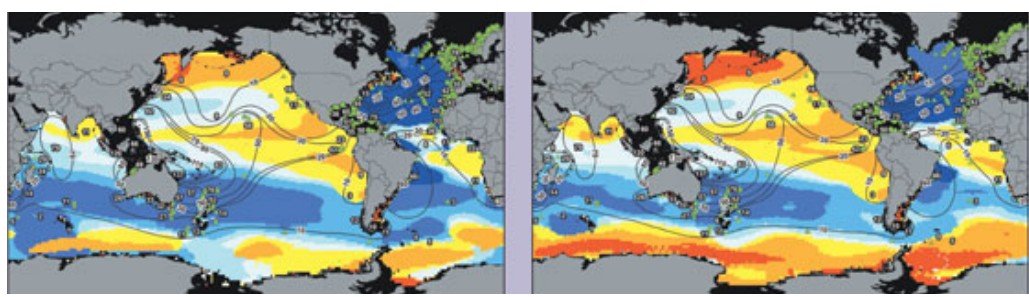

Fig 13. A. Aragonite Saturation Horizon (ASH) B. Aragonite Saturation Horizon (ASH)
       in 2020 with atmospheric CO2 at 440 ppm        2100 with atmospheric CO2 at 788 ppm

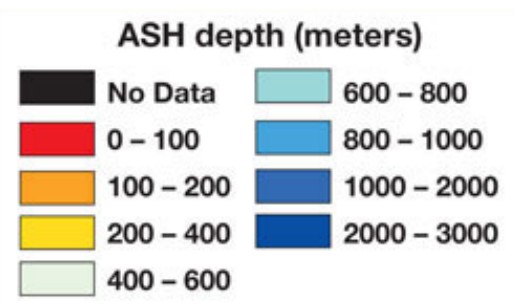

Fig. 12 C. Color Coding for ASH depths.

## 6 HOW RELIABLE ARE pH ESTIMATIONS IN THE SOUTHERN OCEAN USING EMPIRICAL ALGORHITHMS  DERIVED FROM SATTELITE DATA ORGINATING  FROM ARGO FLOTS?

Williams et al. (2016) developed empirical algorithms using high quality Go-SHIP hydrographic
       measurements of parameters such as temperature, salinity, nitrate, pH, pressure and oxygen in
       the Pacific sector of Southern Ocean (Fig. 13).Using coefficients of pH from nitrate and pH from
       oxygen, they generated algorithms that are applied to Southern Ocean Carbon and Climate
       Observations and Modeling (SOCCOM) biogeochemical profiling floats (ARGO-floats) that
incorporate novel sensors to measure pH, nitrate, oxygen, florescence, and back scatters). These
       algorithms were used to estimate pH. (Fig. 14)





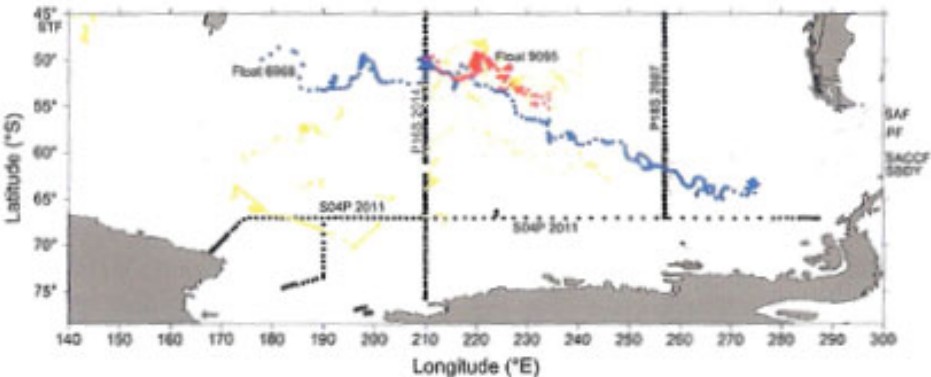

Fig. 14. Map of the trajectories for all SOCCOM profiling floats in the
Pacific sector of the Southern Ocean.

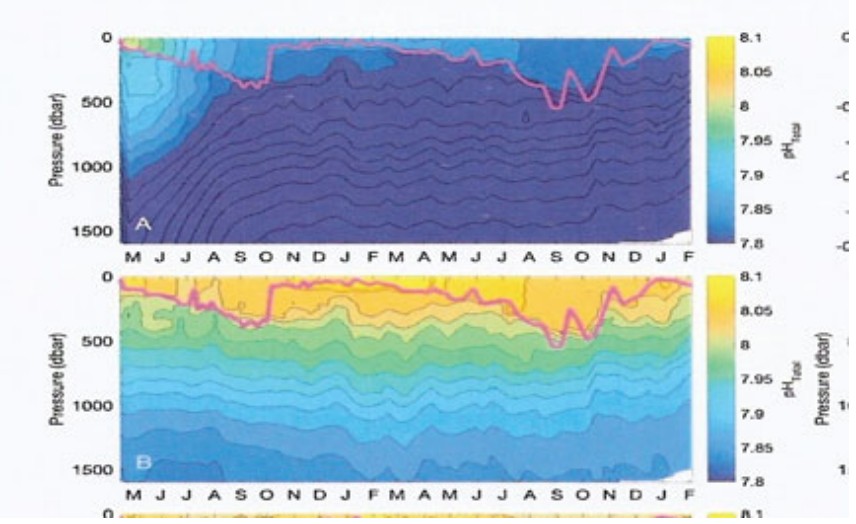

Fig. 15. Measured pH A. CTD water samples and B. Algorithms.

Williams et al. (2016) postulated the view that algorithms presented in their paper for pH
measurements, could be used accurately to estimate pH throughout the full seasonal cycle in
the Southern Ocean. The data suggest apparent change in pH with values of 8 pH in winter vs
15     values of 8.1 pH in austral summers. However, the algorithms do not account for the observed
anthropogenic ocean acidification since it does include any changes in processes such as gas
exchange that may modify the relationship between pH and the master variables used in the
algorithms. Can the 'ARGO' floats, that are not stationary as the moored monitoring arrays
(SOOS), provide data through satellite links to cover the vastness of the Southern Ocean all
20     around Antarctica to predict changing carbonate chemistry in the coming decades? I pose this



question in this paper with the hope we will develop the right technology to keep track of the changes in carbonate chemistry of the seas around Antarctica where krill dwells.

**7 SYNERGISTIC IMPACT OF WARMING AND ACIDIFICATION**

The Antarctic marine ecosystems will be the first ocean regions to become persistently undersaturated with respect to aragonite as a result of anthropogenic-induced acidification. Thus, these ecosystems are natural laboratories in which to test many hypotheses on the
impacts of ocean acidification and other stressors, particularly those induced by global warming.

Using an integrative, experimental approach, Flynn et al. (2015) examined the impacts of near-future warming (−1 (ambient) and 2°C (+3°C)] and ocean acidification [420 (ambient), 650 (moderate) and 1000 µatm $pCO_2$ (high)] on survival, development and metabolic processes over
the course of 3 weeks in early development of Antarctic dragon fish.  In combined warming and ocean acidification scenarios, Antarctic dragon fish embryos experienced a dose-dependent, synergistic decrease in survival and developed more slowly.

It is also important to point out that Talmage and Gobler (2011) evaluated the effects of
elevated temperature and carbon dioxide on growth and survival of larvae and juveniles of three species of Northwestern Atlantic bivalves *Mercenaria mercenaria*, *Crosostrea virginica* and *Aspropecten irridicens* at two temperatures (24 and 28 C) and 3 different CO2 concentrations (representing past, present and future). This study established that increase in temperature and decrease in pH is likely to have negative consequences to these bivalve species.


Despite the warning about consequences of ocean acidification on krill by Kawaguchi et al. (2013), there has not been a single research paper reporting on krill in a high carbon ocean in the past four years! Dr. Kawaguchi and his


collaborators have not followed their report on decline of hatching success in krill in seawater with $pCO_2$ concentrations above 1000 ppm. Evidently, krill eggs exposed to high CO2 during the first three days of embryonic development, significantly reduces development and hatching rates.

Kawaguchi et al. (2013) produced 'circumpolar risk map' of krill hatching success under projected $pCO_2$ levels at the end of the 2100 and 2300, by using the "Representative Concentration Pathway" (RCP) with a high emission scenario and concurrent increase in gas emissions over time leading to high greenhouse gas concentration levels. Krill habitats from the Weddell Sea and Haakon VII Sea were identified as the high-risk areas for krill recruitment at the end of this century. CO₂ emissions must be mitigated much before 2100, and not wait till 2300 to witness the Southern Ocean getting devoid of krill and penguins.

Fig. 16. Percentage of metamorphosed *of Mercenaria mercenaria* larvae grown under three different levels of CO2 and at two different temperatures.






The tolerance to hypoxia conditions by *E. superba* was studied by Tremblay and Able (2015) and it was found that both *E. superba* and *E. pacifica* maintained stable respiration rate down to low critical pO2, values of 6 k Pa 30% (air saturation). *E. superba* was not lethally affected by 6 hrs exposure to moderate hypoxia. Thillart, George and Stromberg (1999) experimentally studied hypoxia tolerance of the arctic krill *Meganyctiphanes norvegica* in Gullmars Fjord in Sweden and discovered that the arctic krill tolerates hypoxia down to 30 % of normal oxygen level but succumbs to hypoxia below this level.

Whiteley (2011) succinctly addressed the potential physiological responses of marine crustaceans to ocean acidification and concluded that marine species at greater risk are those that have limited physiological capacities (therefore narrow metabolic scope) to adjust (or accommodate and acclimate) to environmental change. These vulnerable species are poor iono-osmo regulators and have limited abilities to compensate for acid-base cellular disturbances. The problems are compounded in slow-moving, relatively inactive species because these species have low circulatory protein levels and low buffering capacities. Unlike K-strategist species in deep-sea or high-polar

species*, E.superba* has a high metabolic scope as established by George (1984).

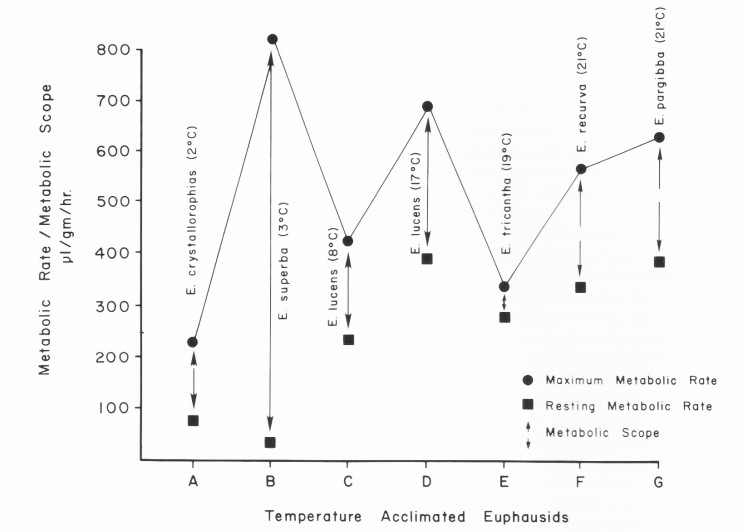

Fig. 17. Metabolic rates and metabolic scope of seven
different species of the genus *Euphausia*.

Antarctic krill *Euphausia superba* (Fig. 18) is agile and active while swimming or feeding and we know that the Antarctic krill forms huge swarms as a "superorganism" for schooling, foraging and anti predatory behavior (Hamner and Hamner, 2000). Krill does not genuinely qualify as the so called "plankton" at the mercy of water movements but krill is a good swimmer. At elevated levels of pCO2 as predicted by Kamwaguchi et al. (2013) in the distant future (year 2300) krill may succumb to OAS stress and will be unable to cope with acidic stress and will fail to go through normal egg development.





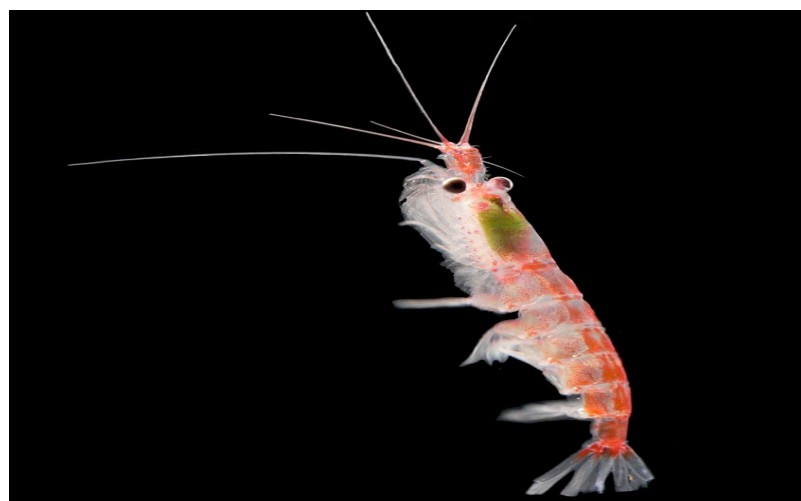

Fig. 18. *Euphausia superba* Dana 1850.

Dupont et al. (2010) hypothesized that species naturally exposed to variable environmental pH conditions may be pre-adapted to future high pCO2 habitats but it is important now to understand and monitor ambient environmental variables in order to be able to predict different sensitivity to future climate change. These authors, on the basis of their experimental work with Echinoderm eggs, embryos and larvae concluded that near-future ocean acidification changes would have a negative impact.

## 8 CONCLUSION

1. At the XXXV Antarctic Treaty Consultative meeting in Buenos Aires from June 20 to July 1, 2011, the following recommendation was made:

"It is imperative that research programs  fill in the gaps of current research on Southern Ocean carbon uptake and one of the main areas currently lacking understanding is related to longer-term studies of acidification on the entire life cycle of important marine species. The work that is underway in the Southern Ocean Observing System (SOOS) will help produce useful information but much more targeted research is required."  Therefore, we need to emphasize on more focused research on impact of ocean acidification on entire life cycle of the Antarctic marine ecosystem keystone *Euphausia superba*.

2. We still do not have sufficient experimental data on embryonic stages, larvae, juvenile and adult krill *E. superba* to reach a conclusion that ocean acidification in the Southern Ocean will pose a threat to krill and krill-based food chain of the Antarctic marine ecosystems. Realistically, we need to do more experiments at pCO2 conditions at different temperatures and habitat pressure (10 to 100 ATM) that we can predict in 2030, 2050 and 2100, rather then in alarming unreal *in situ* conditions as high as 3000 microatoms in 2300, as Kawaguchi et al. (2013) have envisaged in abnormal experimental simulations.



3.  Likewise, we still have to perfect the methods to measure pCO2 at different depths with *in situ* technology, rather using algorithms to calculate pH.  Reliable moorings at critical locations in the Southern Ocean, supplanted by accurate *Argo* float augmentation may be the best approach to obtain accurate predictions of Carbonate chemistry of the Southern Ocean.

**9 ACKNOWLEDGEMENTS**

The research was supported by the US National Science Foundation – Polar Programs under Grant No. DPP-8026535.I wish to thank Capt Peter Lenie of *R/V Hero* for his enthusiastic support to locate krill patches and swarms along the Antarctic Peninsula. Dr. Enrico Marschoff of the Argentine Antarctic Institute helped the author to sample krill swarms off of South Georgia and South Sandwich Islands during the *Islas Orcadas* cruise to the Southern Ocean. Thank you Prof. Linda Hayden of Elizabeth City State University, North Carolina for inspiring me by her collaboration with me on the remote-sensing research, focusing on ice-melting in the Amundson Sea over the past decades. Prof. Stephen Emslie, penguin researcher at UNCW , for critically reading the first draft of the manuscript. This paper is dedicated to 2017 Earth Day.

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
