# Peer review of "Potential Impact of Carbonate Chemistry Change (pCO2) on Krill and Krill-based Food chain in the Southern Ocean with emphasis on Embryogenesis of Antarctic krill."

_Biogeosciences, 2017_

## Referee Comment (RC1) · Anonymous Referee #1 · 7 Aug 2017

This manuscript is a review (no new data presented) with a primary emphasis on the effect of PCO2 on krill in the southern ocean. While the paper expresses some viable concerns, there is just not sufficient data at present to warrant the speculation. There have been only 3 papers published on the effects of ocean acidification on Antarctic krill and the targets range from embryogenesis to adult grazing rates and excretion. The results are highly variable, in some cases preliminary, and not particularly alarming. Only at 2000 ppm PCO2 is embryogenesis affected. Temperature is much more of a concern yet that is hardly touched in this manuscript.

[Figure]

The hypothesis about interacting effects of temperature, hydrostatic pressure and PCO2 on embryogenesis is interesting and worthy of future study. I might support a manuscript focused on this aspect alone without all of the overly-broad review and speculation. A clearly stated hypothesis, some calculations and schematic diagrams are needed.

Krill are not heavily calcified and there is little reason to expect calcification to be an issue for them. The data on crustaceans more broadly is limited and contradictory.

There is no reason to expect hypoxia to be a problem for Antarctic krill yet this manuscript discusses it.

The discussion of pH variability in Antarctica and with depth is important but confusing and poorly written. Krill are exposed to wide variations in PCO2 in swarms, during phytoplankton bloom variation, seasonally etc.

I've never seen PCO2 expressed in microatoms. $\mu$atm.

---

## Referee Comment (RC2) · Anonymous Referee #2 · 6 Sep 2017

This manuscript summarises published studies on ocean acidification impacts on Antarctic krill, and together with the author's previous own work and information from other research in the West Antarctic Peninsula area, this manuscript calls makes statements of areas where future efforts need to be put in.

Unfortunately there is no new data in this manuscript to consist a research article or enough material to consist a standalone review.

The point made by the author regarding the needs for understanding of combined effects of multiple stressors have generally been reviewed in various articles, and Flores et al. (2012) review is a comprehensive on this point specific for Antarctic krill.

The effects of pressure is the only concept that was not covered in Flores et al paper. However, my impression is that effects of temperature is more important over pressure (I could be wrong though), yet its synergistic effects on krill are still not fully investigated.

I see this manuscript as an opinion paper by the author rather than an article or a review paper. Therefore if this manuscript is to be published, my suggestion is to considerably shorten (down to a few pages) and target the manuscript for it to be written as "Ideas and perspectives". This is one of the manuscript categories in Biogeosciences to report "new ideas and novel aspects of scientific investigations" within the journal scope. However, of course, the final call is up to the journal editor.

If the author choose to go along that line the author should only concentrate on the justifications based on the past research and its gaps to draw the 3 conclusions.

---

## Author Comment (AC1) · 8 Sep 2017

I am glad the 2nd anonymous reviewer made this remark "my impression is that the effects of temperature is more important than pressure but I could be wrong." In essence, i wrote the paper to bring to light that pressure can not be ignored and also this paper justifies the need to synergetic effects of pressure and temperature because the high pCo2 conditions are encountered at deeper depths and not at surface. In fact, I am now in the process of seeking NSF funding to do research on synergistic effects fo pressure and temperature.

[Figure]

I hope the journal editor will recognize the timely importance of getting this paper published in BGD.

As far as reviewer number one I concur with the reviewer that this paper realizes the need for exploring the interplay between pressure, temperature and pCO2

---

## Author Comment (AC2) · 4 Oct 2017

Let me inform the editor that agree with the reviewer that there are thus far 3 published papers linking pressure with ocean acidification and my paper wile the fourth one.